# Study on Flow Velocity during Wheeled Capsule Hydraulic Transportation in a Horizontal Pipe

**Yongye Li \*, Yuan Gao, Xihuan Sun and Xuelan Zhang**

College of Water Resource Science and Engineering, Taiyuan University of Technology, Taiyuan 030024, China; miraikoi@126.com (Y.G.); sunxihuan@tyut.edu.cn (X.S.); zhangxuelan11@tyut.edu.cn (X.Z.)

**\*** Correspondence: liyongye@tyut.edu.cn; Tel.: +86-139-3423-9832

**Abstract:** As a clean, low-carbon, and green hydraulic transportation technology, wheeled capsule pipeline hydraulic transportation is a transportation mode conducive to the sustainable development of the social economy. Based on the method of a physical model experiment and hydraulic theory, the flow velocity characteristics in the pipeline when the wheeled capsule with a length–diameter ratio of 2.5 and 2.14, respectively, was transported in the straight pipe section with an inner diameter of 100 mm were studied in this paper. The results show that in the process of transporting materials, the flow velocity distribution of the cross section near the upstream and downstream section of the capsule was basically the same, and the axial velocity was smaller in the middle of the pipe and larger near the inner wall of the pipe. The radial velocity distribution was more thinly spread near the pipe wall and denser near the center of the pipe. The circumferential flow velocity was distributed in the vicinity of the support body of the wheeled capsule. For any annular gap section around the wheeled capsule, the radial velocity of annular gap flow was very small, and the average radial velocity of annular gap flow was about 1/30 of the average axial velocity of annular gap flow and about 0.7 of the average circumferential velocity of annular gap flow. The axial, circumferential, and radial flow velocities on the same radius measuring ring changed with the polar axis in a wave pattern of alternating peaks and troughs. These results can provide the theoretical basis for optimizing structural parameters of the wheeled capsule.

**Keywords:** wheeled capsule; flow velocity; hydraulic transportation; pipeline

---

## 1. Introduction

With the rapid development of economy and society, the transportation industry plays an increasingly important role in economic development [1,2]. However, at present, the crude development mode of energy is still adopted in the field of modern transportation, which leads to a large amount of fossil fuel consumption and greenhouse gas emissions. At the same time, with the continuous expansion of the economy, the consumption structure is also constantly upgraded, industrialization is rapidly promoted, human consumption of resources is also increasing, and the contradiction between the carrying capacity of transportation and the total demand of society is also expanding. Railway transportation, highway transportation, air transportation, and waterway transportation no longer meet the total social demand of rapid development. Urban traffic congestion, traditional transportation, serious environmental pollution, and a logistics bottleneck of emerging e-commerce are all primary problems troubling cities. Facing the increasingly severe constraints of resources and environment, it is necessary to establish a sense of survival crisis and the development concept of environmental protection and low-carbon transport, make energy conservation and emission reduction the top priority, focus on energy conservation in water conservancy, power, transportation, and construction, adjust the energy consumption structure, and increase the proportion of nonfossil

energy. Therefore, the development of clean, low-carbon, and green new modes of transportation and related technologies has become an important research topic in the field of transportation in the future. Pipeline transportation is one of the energy-saving and environment-friendly transportation technologies based on this concept. It is a transportation mode conducive to the sustainable development of the national economy and has broad application prospects.

After several years of development, pipeline transportation has gradually changed from low concentration, small diameter, and short distance to high concentration, large diameter, and long distance [3,4]. As a branch of pipeline transportation, pipeline hydraulic transportation mainly includes slurry pipeline hydraulic transportation, mold pipeline hydraulic transportation, and capsule pipeline hydraulic transportation. Slurry pipeline hydraulic transportation [5,6] is a transportation mold that mixes granular solid material with liquid transportation medium to make slurry, which is transported by pumping in the pipeline and separated at the destination. The mold pipeline hydraulic transportation [7,8] is a transportation mold that the bulk material is molded or extruded into cylindrical material resistant to water and abrasion, and then the mold material is injected into the transporting pipe to carry out long-distance transportation with pump as the power and water as the carrier. To some extent, the above-mentioned two kinds of pipeline hydraulic transportation technologies alleviate the traffic pressure and environmental pollution on the ground, but both of them have the defect that only solid materials can be transported [9,10]. The capsule pipeline hydraulic transportation [11] is another kind of pipeline hydraulic transportation technology after the slurry pipeline hydraulic transportation and the mold pipeline hydraulic transportation. It is a solid–liquid separation of transportation mode in which the material (liquid or solid) is packed and sealed in the capsule and then injected into the conveying pipe to promote the movement of the capsule by flow.

Many scholars have studied the transportation characteristics of the capsule pipeline hydraulic transportation since it was put forward. Liu et al. [12,13] studied the hydraulic characteristics of the flow in the pipeline when the capsule was still and obtained the law of the pressure change of the flow in the pipeline based on the theory of pipeline hydraulics. Tomita et al. [14] analyzed the velocity characteristics of the capsule in the state of hydraulic balance, calculated the influence of the length–diameter ratio and friction coefficient of the capsule on the velocity of the capsule, and put forward feasible suggestions on the problem of pipeline drag reduction based on the theory of concentric annular gap flow. Kruyer et al. [15] studied the velocity and pressure drop of the transport medium when the diameter ratio of the capsule to the pipe was between 0.25 and 0.97 in the two states of the concentric and eccentric of the capsule. It was concluded that the larger the ratio of the diameter, the smaller the velocity of the transport medium. Ginevskii et al. [16] verified that the transport speed of the capsule in the turbulent state can reach 1.5 times the velocity of the fluid medium, but it cannot be achieved in the laminar state by experiments. Taimoor et al. [17–19] studied the movement velocity of rectangular, cylindrical, and spherical capsules, analyzed the relationship between the geometry of the capsule and the flow pressure in the pipeline, and derived the semiempirical formula for calculating the flow pressure drop in the pipeline when the capsule moved in the pressurized pipe. Deniz et al. [20] studied the pressure distribution of flow in the pipeline when the capsule moved under the conditions of different flow discharges, different density of the capsule, and different elbow angle and determined the pressure gradient of flow in the pipeline gradually increased with the increase of flow discharge, density of the capsule, and elbow angle. Li et al. [21–23] studied the movement characteristics of the capsule under different length diameter ratio, flow discharge, and transporting load conditions, established the movement model of the capsule under various influencing factors, and analyzed the transporting energy consumption of the capsule in the process of transporting materials. Tong et al. [24] optimized the system of the capsule pipeline hydraulic transportation from the aspects of power and regulation system, delivery and receiving device, test pipeline, and test device. Zhang et al. [25,26] studied the pressure distribution of the cross section of the pipeline during the movement of capsule and analyzed the relationship between the pressure distribution characteristics of the annular gap flow generated by the movement of the capsule in the

pipeline and the flow discharge. Zhang et al. [27,28] studied the pressure characteristics of annular gap flow when the capsule was still in the pipeline and concluded that the pressure of annular gap flow first decreased, then increased, and then decreased from the inlet to the outlet. However, as one of the core components of the capsule pipeline hydraulic transportation, the different structure of the capsule would affect the flow in the pipeline during the material transport process of the capsule, thus affecting its movement stability. Therefore, in this study, the flow velocity characteristics during the wheeled capsule hydraulic transportation in straight pipe section were investigated aiming to provide theoretical basis for optimizing capsule structure parameters.

## 2. Experimental Design

In this study, the wheeled capsule [29,30] is mainly composed of the barrel, the support body, and the universal ball. The barrel is a hollow cylinder that can transport not only solid materials but also liquid and gas materials. Three cylindrical support bodies of 120° interval angle were installed at both ends of the barrel, respectively. The existence of the support bodies made the center of the capsule always coincide with the central axis of the pipeline during the movement of the capsule and kept their movement concentric, thus overcoming the defect of unstable motion of the barrel. Universal balls were installed separately at the end of each support body. The existence of the universal balls changed sliding friction into rolling friction during the movement of the capsule, thus prolonging the service life of the capsule and the pipe. The structural diagram of the wheeled capsule was shown in Figure 1.

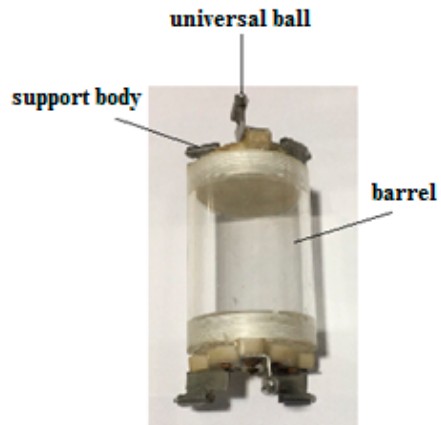

**Figure 1.** Structural diagram of wheeled capsule.

The experimental device [29–31] was mainly composed of a power device, a capsule delivery and receiving device, and experimental pipe. During the experiment, water was pumped out of the underground reservoir by centrifugal pump and flowed into the plexiglass pipe through the steel pipe. The capsule was placed into the test pipe from the delivery device and was secured by the brake device. We adjusted flow discharge through the gate valve to the flow discharge required by the test. The flow discharge was measured by turbine flow meter. After the flow was stable, we removed the braking device, released the capsule, and measured the flow velocity by phase doppler analyzer (PDA) in the pipeline during the movement of the capsule in the test section. In order to prevent the refraction of the pipeline surface to the laser, a rectangular water jacket was installed outside the test pipeline. The particle size of the tracer used in the experiment was 30 μ m. Finally, the capsule entered the receiving device from the pipe outlet, and the water flowed into the underground reservoir through the outlet pool, forming a circulation system. The schematic diagram of the test device was shown in Figure 2. In this study, the length–diameter ratio of the capsule was 2.5 and 2.14, respectively, and the flow discharge was 40 m$^3$/h.

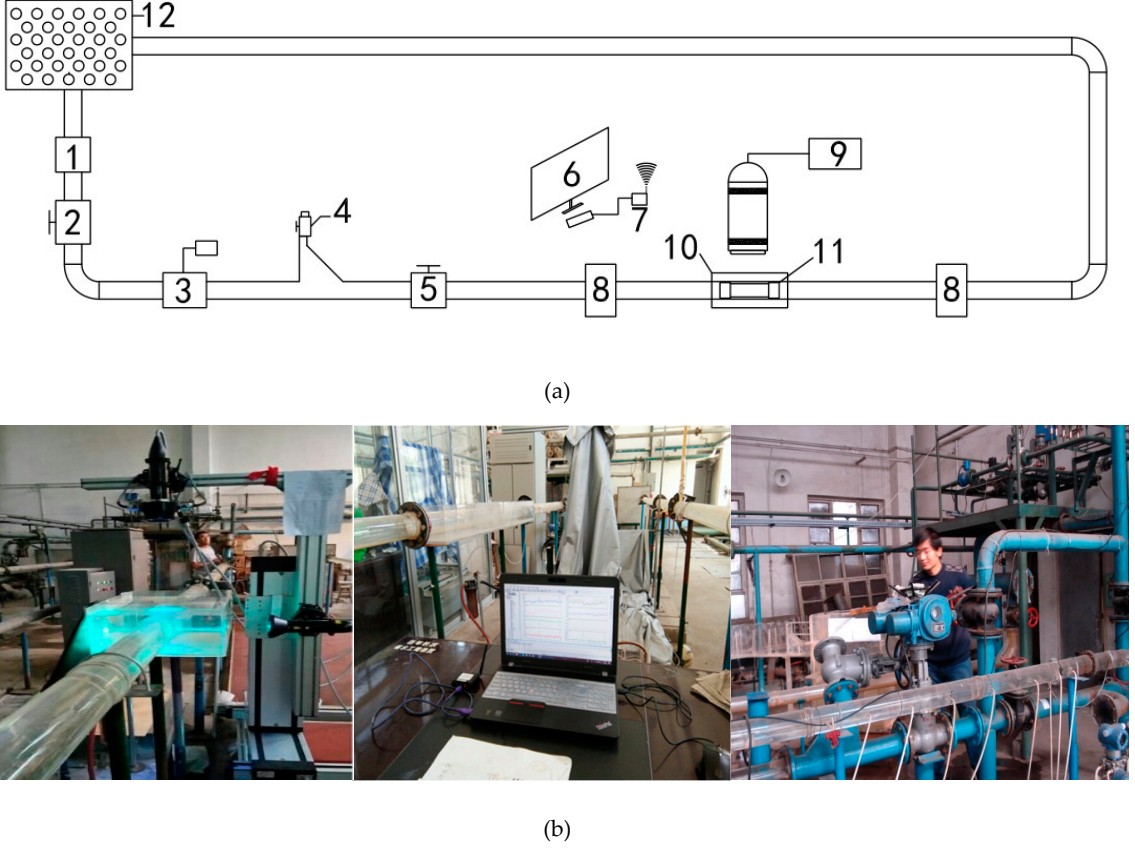

(a)

(b)

**Figure 2.** Experimental system device: (**a**) schematic diagram: 1. centrifugal pump; 2. regulating valve; 3. turbine flow meter; 4. the capsule delivery device; 5. brake device; 6. computer; 7. flow velocity receiving device; 8. the photoelectric sensor; 9. phase doppler analyzer (PDA); 10. rectangular water jacket; 11. capsule; and 12. the water supply device and the capsule receiving device. (**b**) Physical diagram.

The movement of the capsule in the pipe was related to the flow characteristics of the water in the pipe. In the process of transporting materials, the power of the movement of the capsule mainly came from the pressure difference between the front and rear end faces of the capsule, and the resistance of the movement of the capsule was mainly the friction between the pipe wall and the capsule and the friction between the surface of the capsule and the flow. When the flow velocity in the pipe was less than the starting flow velocity of the capsule, the power acting on the capsule was less than the resistance and the capsule was at rest. When the flow velocity in the pipe was equal to the starting flow velocity of the capsule, the power acting on the capsule was equal to the resistance and the capsule was in the critical starting state. When the flow velocity in the pipe was greater than the starting flow velocity of the capsule, the power acting on the capsule was greater than the resistance and the capsule was in an accelerated state and then reached a stable state. The power acting on the capsule overcame the resistance to do work to push it to move along the pipe to transport materials. The flow velocity characteristics in straight pipe were mainly studied during the stable movement of the capsule in this paper.

The flow velocity measurement of the test section can be divided into two situations: one was the flow velocity measurement of the upstream and downstream section of the capsule during the movement of the capsule; the other was the flow velocity measurement of the annular gap section between the outer diameter of the capsule and the inner diameter of the pipeline during the movement of the capsule. In both cases, polar coordinates and measuring rings were used to arrange measuring points, and 12 polar axes and 5 measuring rings were arranged in total. The intersection of the polar axis and the measuring ring was the measuring point. The arrangement of measuring points was

shown in Figure 3. The division of annular gap section of the capsule was shown in Figure 4. In this paper, along the flow direction, the upstream of the capsule was defined as the rear section, and the downstream was defined as the front section.

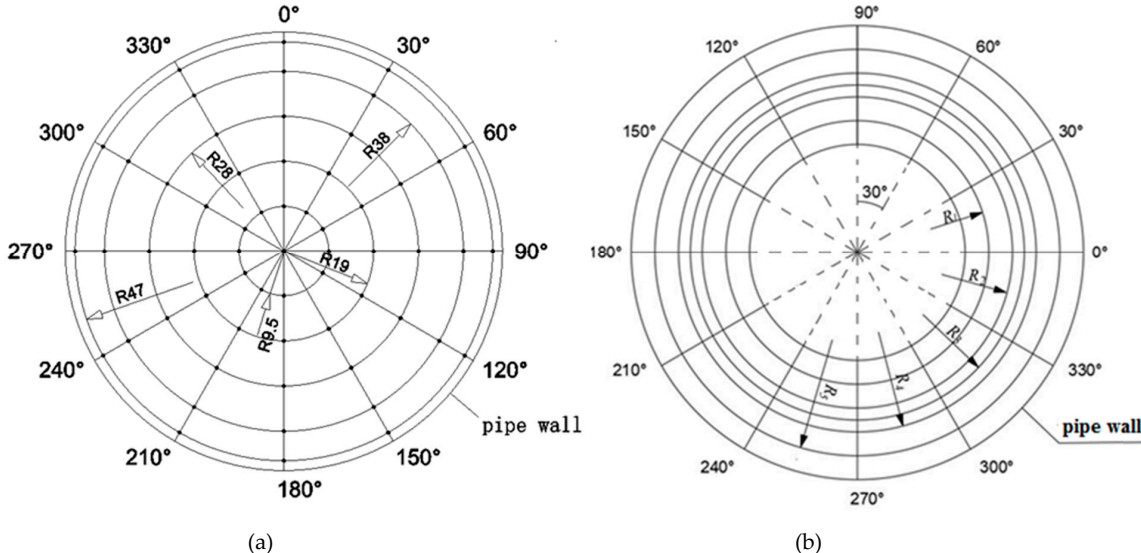

(a)                                                                 (b)

**Figure 3.** Layout of measuring points of sections: (**a**) the front and rear sections of capsule and (**b**) the annular gap section.

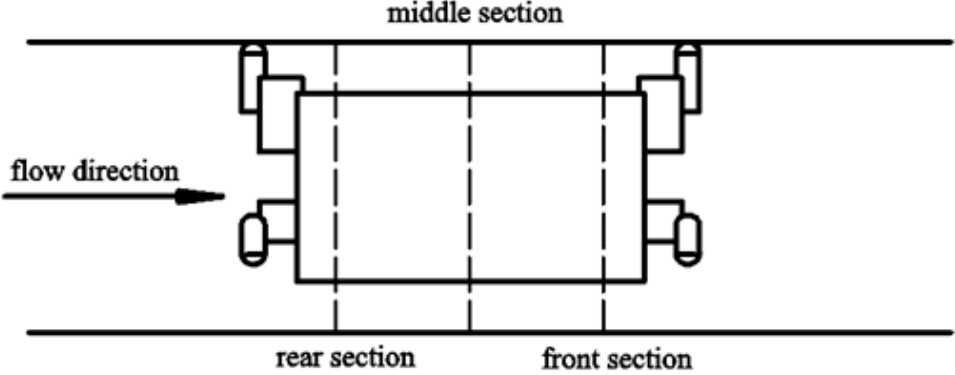

**Figure 4.** Schematic diagram of annular gap section division of capsule.

## 3. Results and Discussion

In the process of transporting materials, the existence of the capsule would lead to the phenomenon of circumfluence, and the flow between the capsule and the wall was concentric annular gap flow, which all changed the velocity distribution of the original flow in the pipeline. The flow velocity distribution of the upstream and downstream sections of the capsule and the annular gap sections between the capsule and the pipe and the influence range of the capsule on the flow were studied based on the fundamentals of two-fluid dynamics and the theory of hydraulics [32,33].

### 3.1. The Flow Velocity Distribution of The Upstream Section of the Capsule

Taking $Z = 0.1$ m and $Z = 3$ m away from the upstream sections of the capsule as examples, the flow velocity distribution of the upstream sections of the capsule during material transportation was analyzed.

Figure 5 shows the distributions of axial flow velocity of the upstream sections of the capsules, as follows:

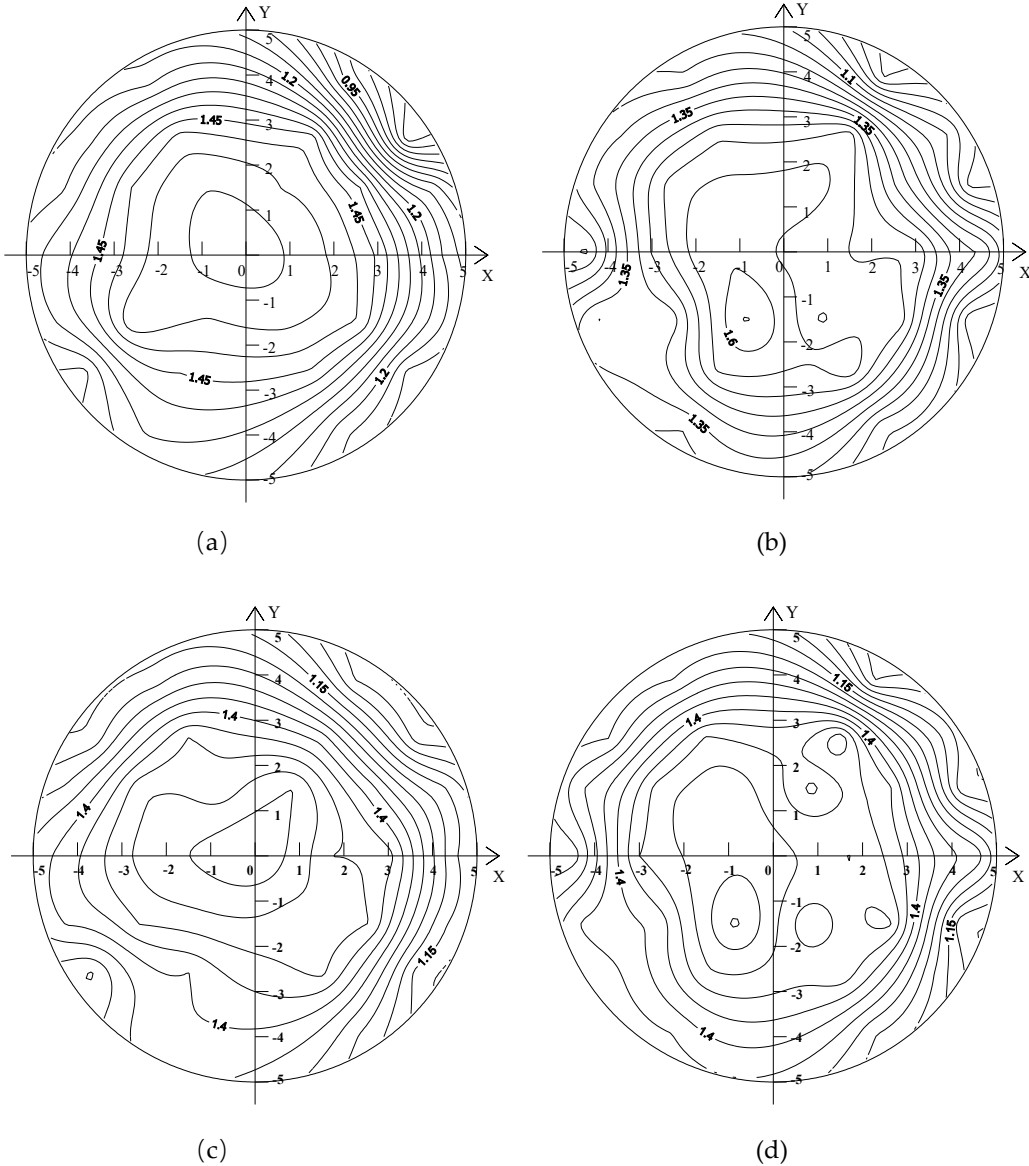

**Figure 5.** Axial velocity distributions of the upstream sections of capsules: (**a**) *L*/*d* = 2.5, *Z* = 3 m; (**b**) *L*/*d* = 2.14, *Z* = 3 m; (**c**) *L*/*d* = 2.5, *Z* = 0.1 m; and (**d**) *L*/*d* = 2.14, *Z* = 0.1 m.

The axial velocity distribution of the upstream section of the capsule was basically similar, and there were several rings with the center of the pipe section as the center. The axial velocity of the upstream section of the capsule was smaller in the middle area of the pipeline and larger near the inner wall of the pipeline. The farther it was away from the center of the pipeline, the greater the change of the axial velocity gradient was. The distribution of axial velocity in the figure was analyzed according to four coordinate quadrants. It can be seen that the distribution of axial velocity was basically symmetric along the direction of the Y-axis, but along the direction of the X-axis, the distribution of axial velocity was different. The axial velocity of the lower section of the X-axis was larger than that of the upper section. For the capsule with different length–diameter ratios, the axial velocity of their upstream sections was evenly distributed in the range of the Y-axis coordinate (3,5), and the velocity gradient changed less. Meanwhile, the smaller the pole radius of the measuring ring was, the greater the corresponding axial velocity was.

Figure 6 shows the distributions of circumferential flow velocity of the upstream sections of the capsules, as follows:

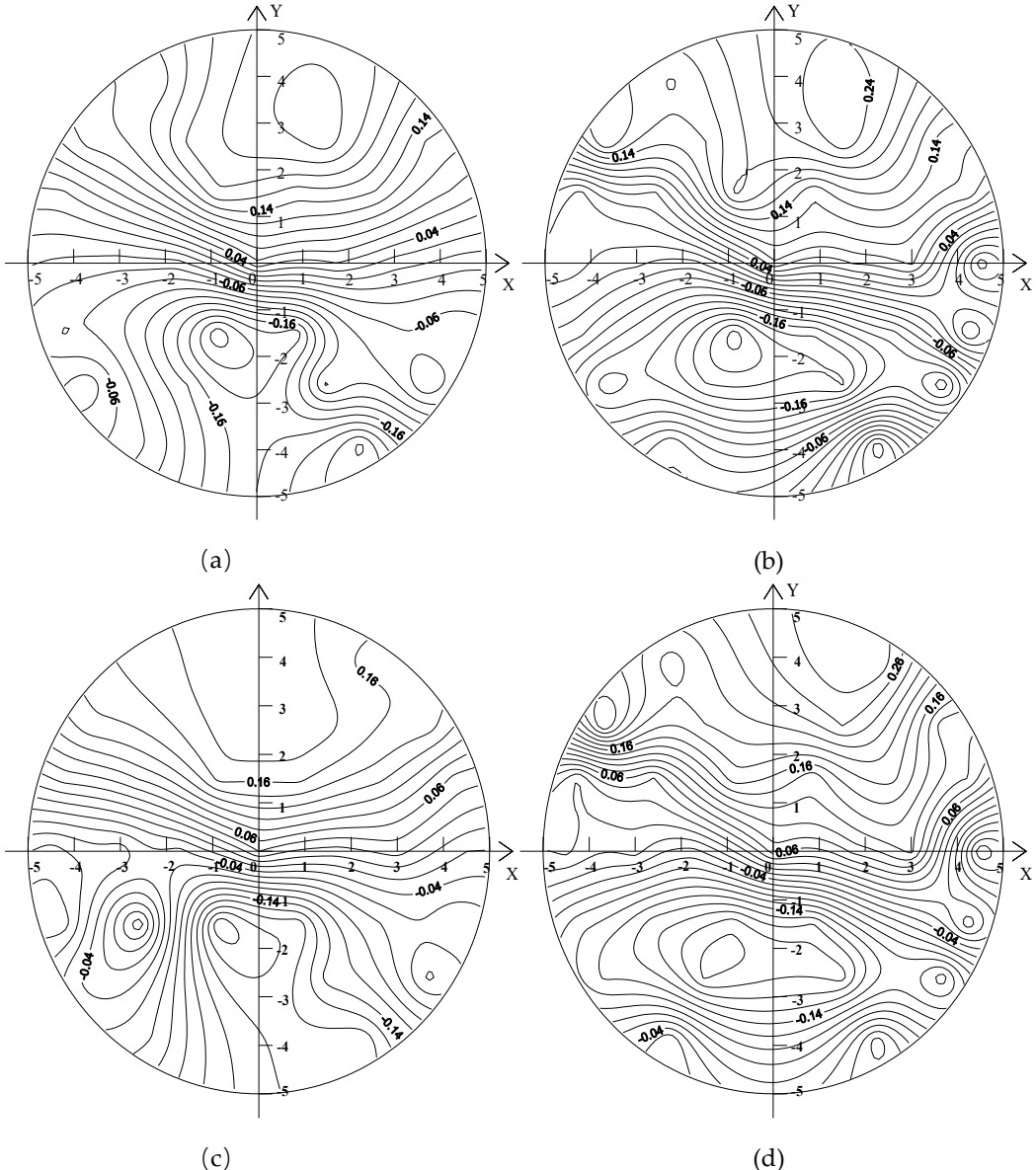

**Figure 6.** Circumferential velocity distributions of the upstream sections of capsules: (**a**) *L/d* = 2.5, *Z* = 3 m; (**b**) *L/d* = 2.14, *Z* = 3 m; (**c**) L/d = 2.5, Z = 0.1 m; and (**d**) *L/d* = 2.14, Z = 0.1 m.

Regardless of the transport condition, the circumferential flow velocity of the upstream section of the capsule changed within the range of (−0.02, 0.2), and the distribution law was basically the same. Above the X-axis, the circumferential velocity was positive, indicating that the direction of circumferential velocity is counterclockwise along the tangent line of the concentric circle, while it was opposite below the X-axis. The circumferential velocity on the X-axis was the smallest and fluctuated up and down the X-axis with the X-axis as the axis of symmetry. The circumferential velocity of each measuring point in the upstream section of the capsule with the length–diameter ratio of 2.5 changed little, while the circumferential velocity of each measuring point in the upstream section of the capsule with the length–diameter ratio of 2.14 changed obviously, indicating that the length–diameter ratio of the capsule has a certain influence on the circumferential velocity of the upstream section of the capsule. However, regardless of the transport condition, the fluctuation of the circumferential velocity at the measuring point (0, 0°) located in the center of the test section was the least, which was less affected by the length–diameter ratio of the capsule.

Figure 7 shows the distributions of radial flow velocity of the upstream sections of the capsules, as follows:

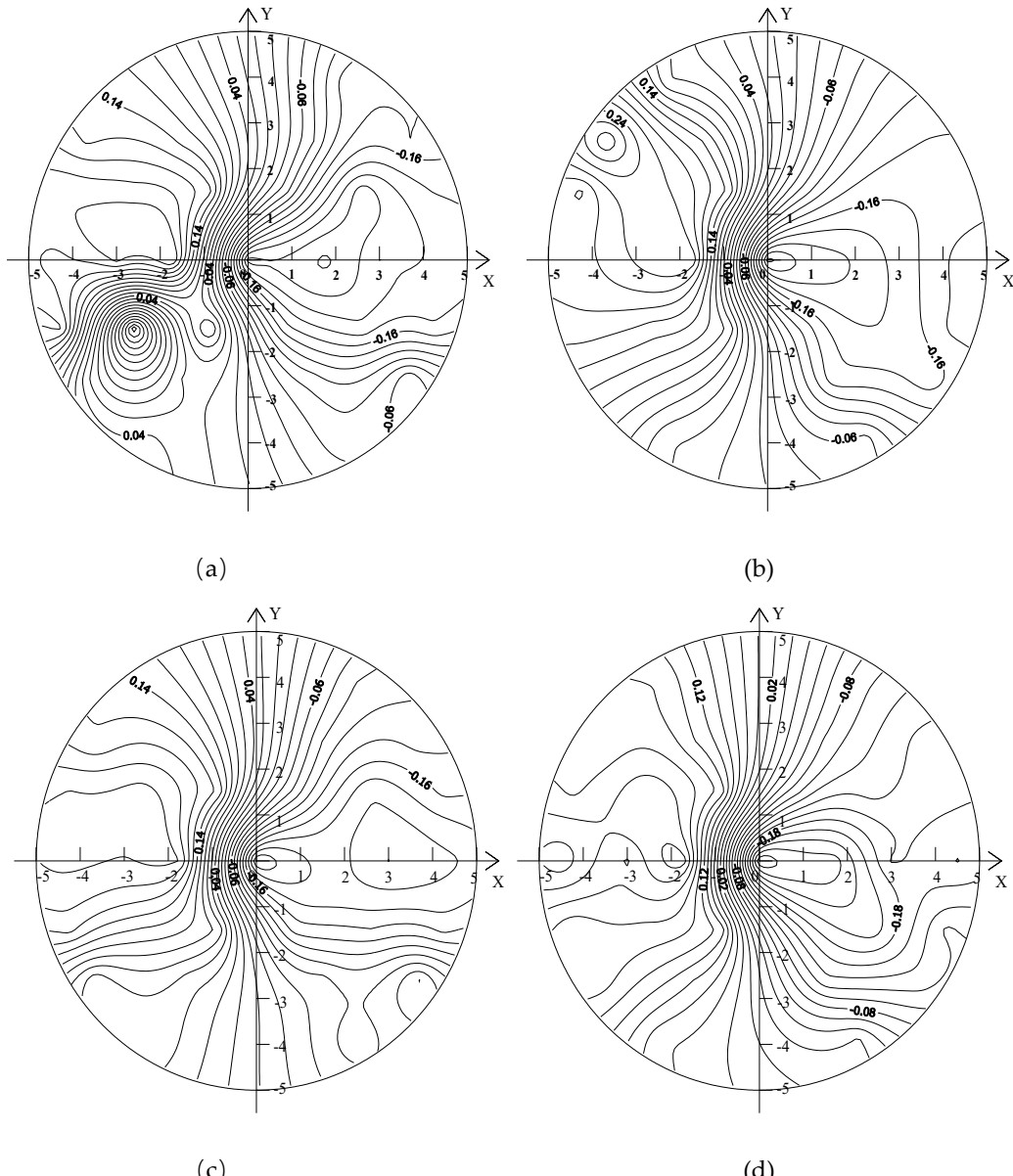

**Figure 7.** Radial velocity distributions of the upstream sections of capsules: (**a**) $L/d$ = 2.5, $Z$ = 3 m; (**b**) $L/d$ = 2.14, $Z$ = 3 m; (**c**) $L/d$ = 2.5, $Z$ = 0.1 m; and (**d**) $L/d$ = 2.14, $Z$ = 0.1 m.

Regardless of the transport condition, the radial velocity distribution of the upstream section of the capsule was more thinly spread near the pipe wall and denser near the center of the pipe. The radial velocity changed in the range of (−0.02, 0.2). The radial velocity on the left side of the Y-axis was positive, while the radial velocity on the right side of the Y-axis was negative. The radial velocity on the Y-axis was the smallest and fluctuated around the Y-axis with the Y-axis as the axis of symmetry. The distribution of radial velocity in the figure was analyzed according to four coordinate quadrants, and it was found that due to the influence of the support body of the capsule, the locations with large disturbance of radial velocity appeared in the third quadrant.

### 3.2. The Flow Velocity Distribution of the Downstream Section of the Capsule

Taking $Z = 0.2$ m and $Z = 3$ m away from the downstream sections of the capsule as examples, the flow velocity distribution of the downstream sections of the capsule during material transportation was analyzed.

Figure 8 shows the distributions of axial flow velocity of the downstream sections of the capsules, as follows:

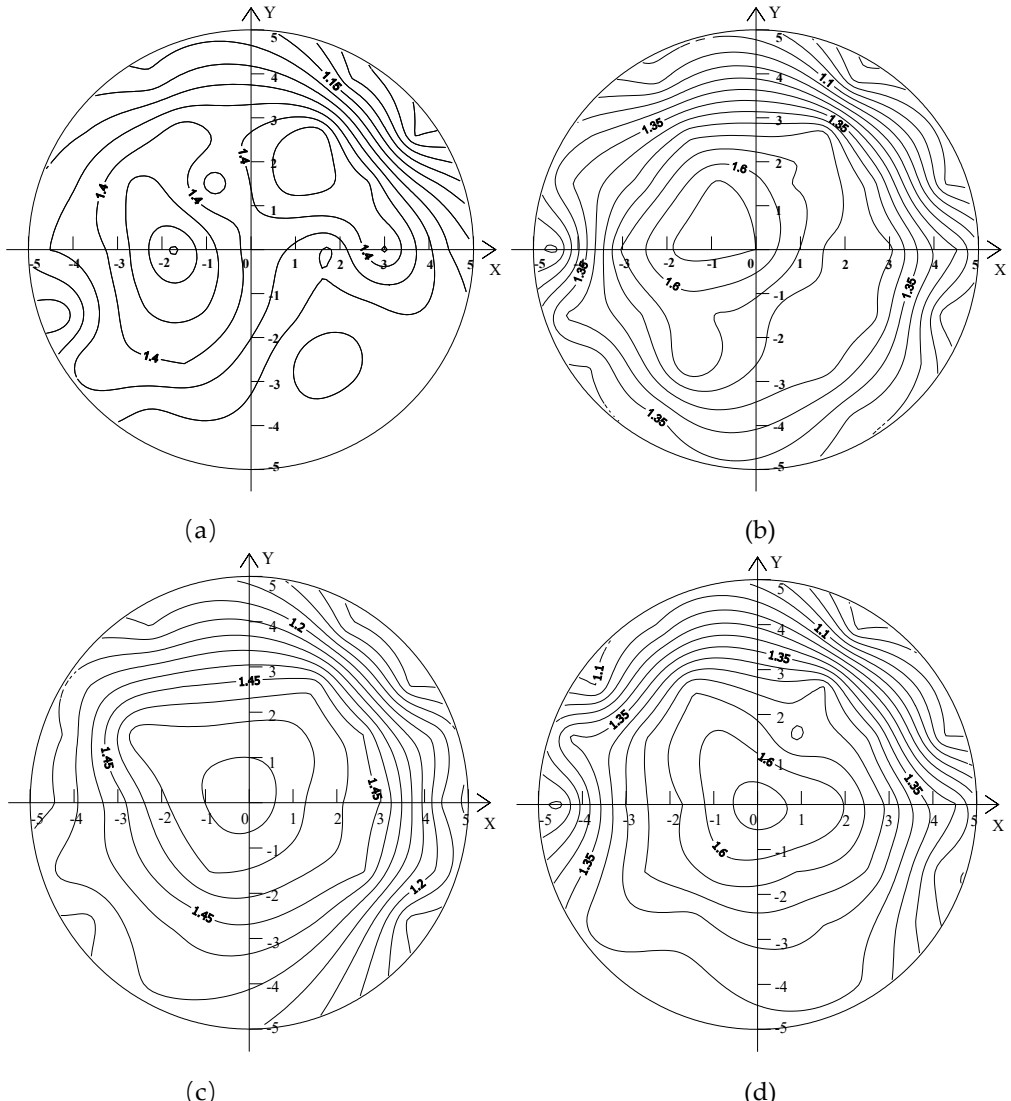

(a)

(b)

(c)

(d)

**Figure 8.** Axial velocity distributions of the downstream sections of capsules: (**a**) $L/d = 2.5$, $Z = 0.2$ m; (**b**) $L/d = 2.14$, $Z = 0.2$ m; (**c**) $L/d = 2.5$, $Z = 3$ m; and (**d**) $L/d = 2.14$, $Z = 3$ m.

As a whole, the axial velocity of the downstream section of the capsule fluctuated within the range of (0.8,1.6). For the same capsule, the distribution of the axial velocity contour in the sections $Z = 0.2$ m and $Z = 3$ m away from the downstream of the capsule was denser than that in the upstream of the capsule, and the axial velocity distribution no longer presented concentric circle distribution but appeared as many small rings. The axial velocity at the measuring point $(0, 0°)$ was the most obvious at the section $Z = 0.2$ m away from the downstream of the capsule. When $L/d = 2.5$, the axial velocity at the measuring point decreased the most.

Figure 9 shows the distributions of circumferential flow velocity of the downstream sections of the capsules, as follows:

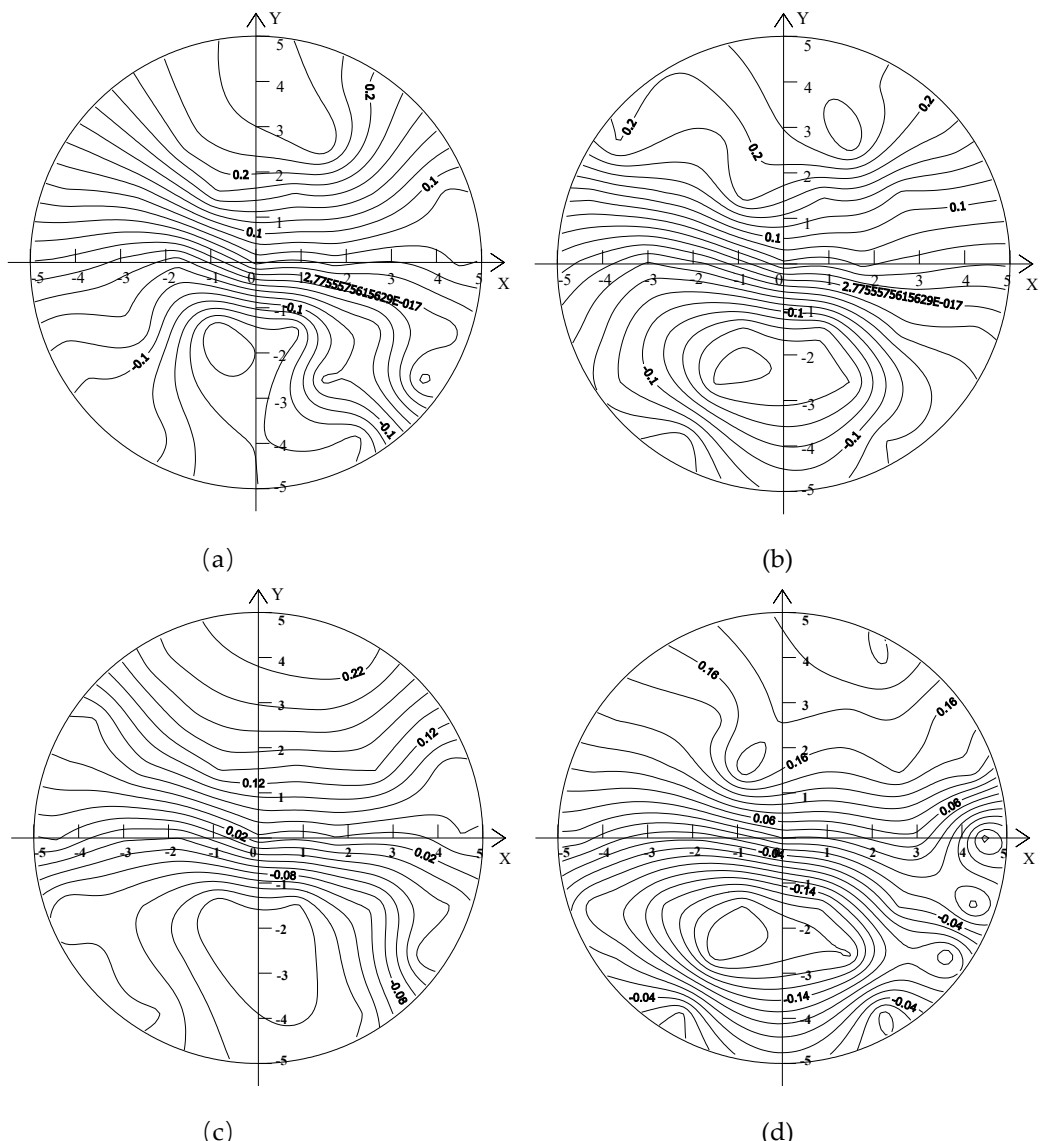

**Figure 9.** Circumferential velocity distribution of the downstream sections of capsules: (**a**) *L/d* = 2.5, *Z* = 0.2 m; (**b**) *L/d* = 2.14, *Z* = 0.2 m; (**c**) *L/d* = 2.5, *Z* = 3 m; and (**d**) *L/d* = 2.14, *Z* = 3 m.

Regardless of the transport condition, the circumferential velocity of the downstream section of the capsule was smaller and the distribution of velocity was relatively stable. Moreover, the farther the downstream section was from the capsule, the smaller the circumferential velocity was. Under the condition of the same flow discharge and transporting loading, the distribution of the circumferential velocity for the same section downstream of the capsule was mainly influenced by the size of the capsule structure. It can be seen from Figure 1 that the capsule is mainly composed of the barrel, the support body, and the universal ball, in which the universal balls were embedded in the support body. It can be seen that the size of the barrel and the support body affected directly the circumferential velocity distribution of the downstream section of the capsule, in which the support body had a greater impact on the circumferential velocity. However, the size of the support body of the capsule studied in this paper is the same. Therefore, for the two capsules of the length–diameter ratio selected in this paper, the proportion of the volume of support body in the volume of capsule with *L/d* = 2.5 was the largest and the smallest in the capsule with *L/d* = 2.14. It can be seen that the capsule with *L/d* = 2.5 had a greater impact on the circumferential velocity of the downstream section, while the capsule with *L/d* = 2.14 had a smaller impact on the circumferential velocity of the downstream section.

Figure 10 shows the distributions of radial flow velocity of the downstream sections of the capsules, as follows:

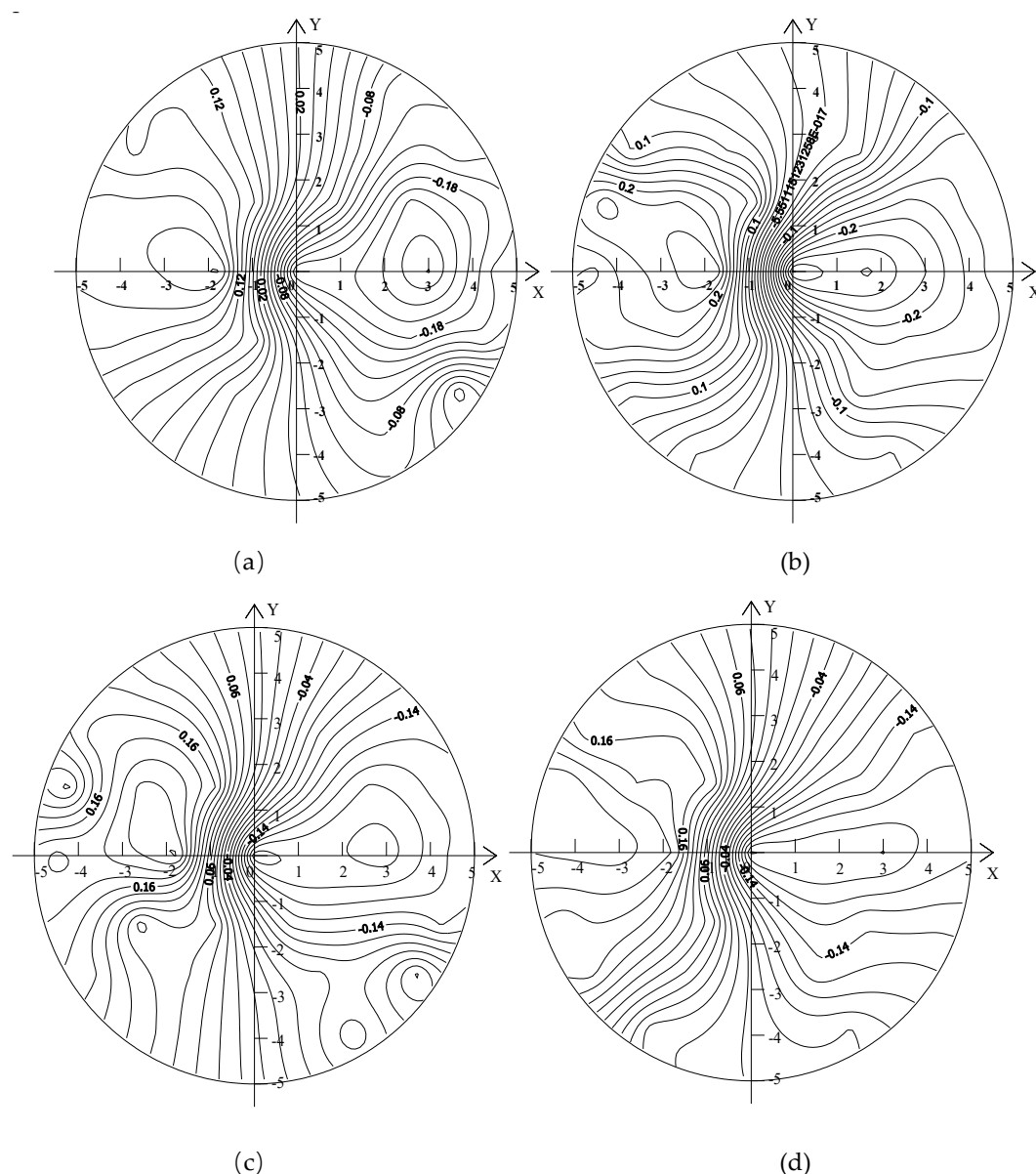

**Figure 10.** Radial velocity distribution of the downstream sections of capsules: (**a**) *L/d* = 2.5, *Z* = 0.2 m; (**b**) *L/d* = 2.14, *Z* = 0.2 m; (**c**) *L/d* = 2.5, *Z* = 3 m; and (**d**) *L/d* = 2.14, *Z* = 3 m.

Regardless of the transport condition, the radial velocity distribution of the downstream section of the capsule was basically the same. The radial velocity distribution of the downstream section of the capsule is more thinly spread near the pipe wall and denser near the center of the pipe, but its value was always small. The farther the downstream section was from the capsule, the smaller the radial velocity was. When the length or the diameter of the capsule was the same, the radial velocity of the same measuring point increased with the increase of the length–diameter ratio of the capsule, but the increase was not significant.

### 3.3. Flow Velocity Characteristics of Annular Gap Section Between Capsule and Pipe

When the capsule was running in the pipeline, the annular gap flow with a certain length was formed along the length direction of the capsule. The flow velocity distribution of the annular gap

section of the capsule in the process of transporting materials was analyzed. The front, middle, and rear sections of the capsule were selected for study. The division of the front, middle, and rear section of the capsule was shown in Figure 4.

Figure 11 shows the axial flow velocity distribution of the front, middle, and rear annular gap sections of the capsule, as follows:

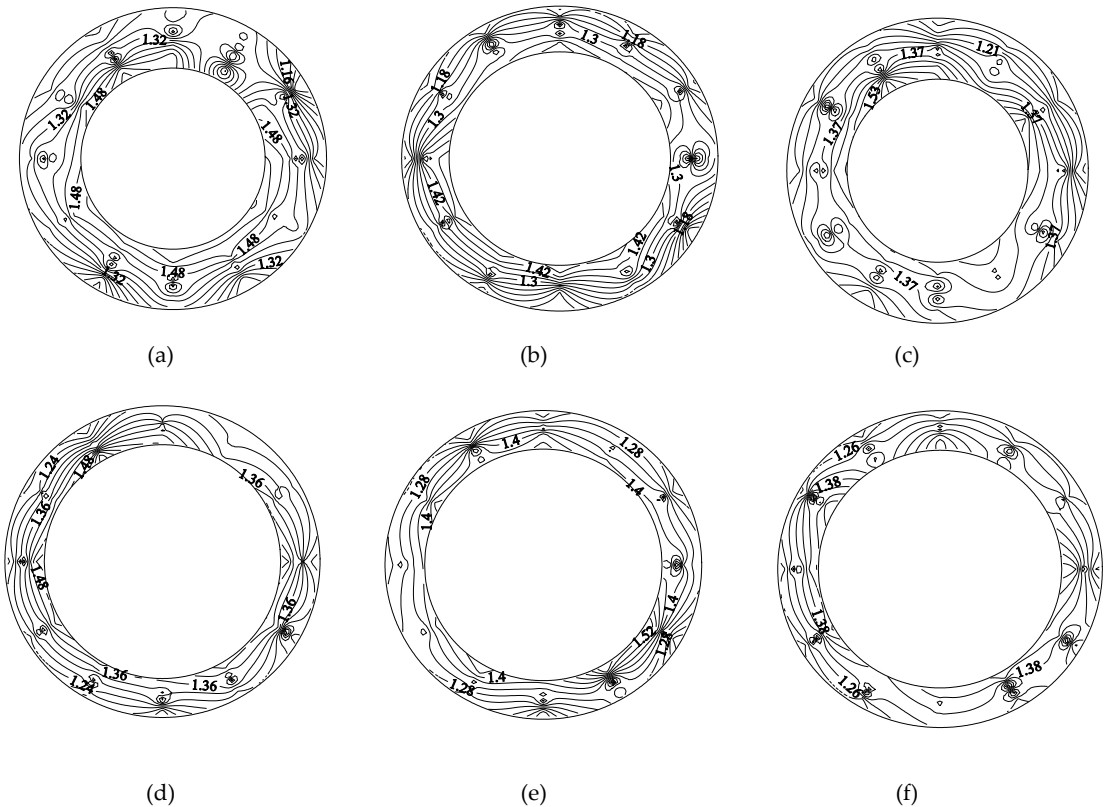

(a)                                             (b)                                            (c)

(d)                                             (e)                                            (f)

**Figure 11.** Axial flow velocity distribution of annular gap sections of the capsules: (**a**) $L/d$ = 2.5, rear section; (**b**) $L/d$ = 2.5, middle section; (**c**) $L/d$ = 2.5, front section; (**d**) $L/d$ = 2.14, rear section; (**e**) $L/d$ = 2.14, middle section; and (**f**) $L/d$ = 2.14, front section.

1) Along the movement direction of the capsule, the axial velocity contour line of annular gap flow changed from dense to sparse, indicating that the axial velocity gradient of annular gap flow decreased along the length direction of the capsule, and its distribution gradually became uniform.

2) When the water flowed around the rear section of the capsule, the axial velocity distribution of the rear section of the capsule was more disordered, and the contour line was denser for the double influence of the support body and the barrel. Its axial velocity was larger than that of the middle section and the front section. When the water continued to flow to the middle section of the capsule, the axial velocity distribution was flatter than that of the rear section of the capsule, and the density of contour line was also lower than that of the rear section of the capsule. Its axial velocity distribution began to appear in the high speed and low speed regions gradually. When the flow continued to the front section of the capsule, the axial velocity distribution was flatter and the contour lines were sparser than that of the rear section and the middle section. Moreover, because of the influence of the cylindrical support body, six distribution regions of axial velocity appeared, including 0°, 120°, and 240° for the high speed distribution and 60°, 180°, and 300° for the low speed distribution.

In order to further analyze the axial velocity change of annular gap flow during the movement of the capsule, the capsule with $L/d$ = 2.14 as an example was analyzed from the same polar axis and the same measuring ring. The arrangement of the polar axis and the measuring ring was shown in Figure 3.

Figure 12 shows the local variation of axial velocity on the same polar axis.

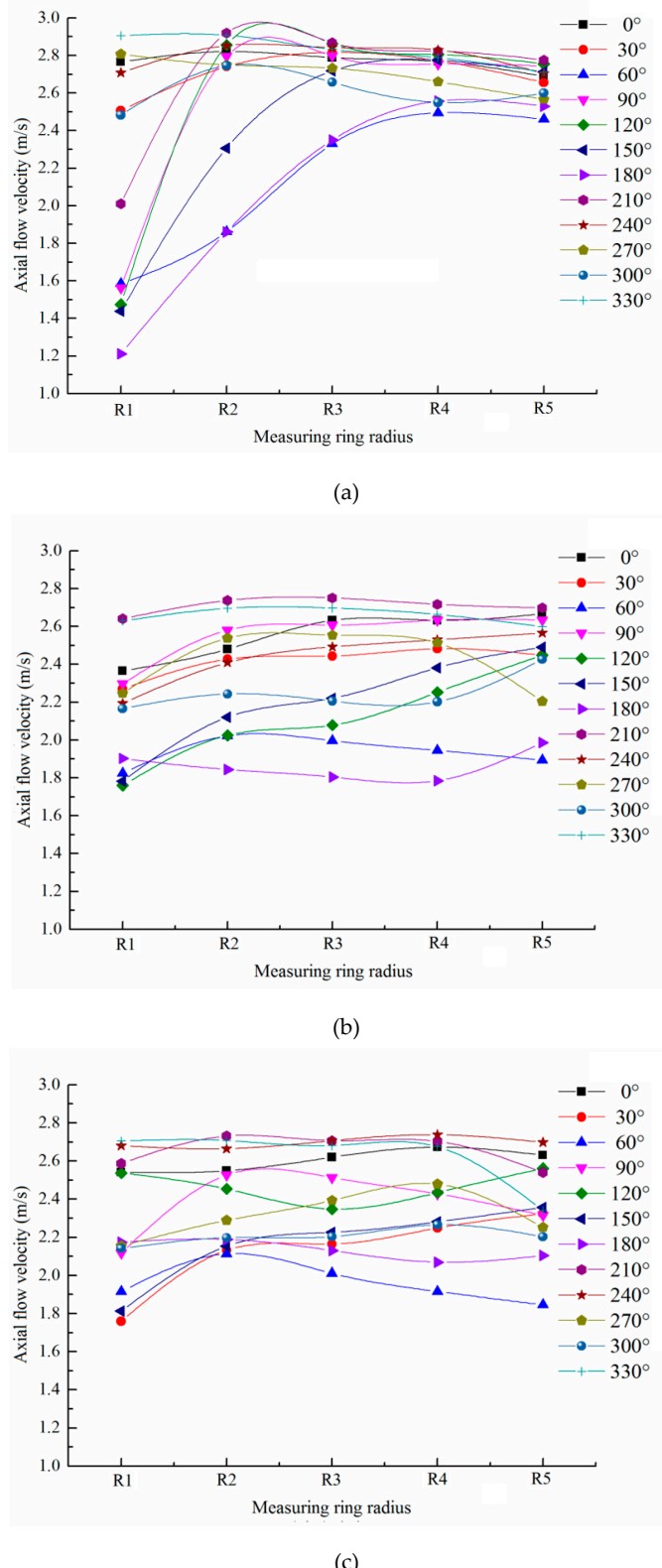

**Figure 12.** Axial velocity of the same polar axis: (**a**) rear section; (**b**) middle section; and (**c**) front section.

For the rear section, the axial velocity on each pole axis increased first, then decreases, and then tended to be uniform with the increase of the ring radius. The axial velocity distribution on the maximum radius ring R5 was in the range (2.4 m/s, 2.8 m/s), and the axial velocity distribution on the minimum radius ring R1 was in the range (1.2 m/s, 2.9 m/s). When the flow around the tail section

of the capsule flowed forward along the annular gap area, the flow velocity would be redistributed. The axial velocity distribution interval on the R1 measuring ring in the central section of the capsule had been reduced to (1.6 m/s, 2.6 m/s), while the axial velocity distribution interval on the R1 measuring ring had also been reduced to basically the same as that on the R1 measuring ring. When the water continued to flow through the front section of the capsule, the variation range of the axial velocity between the poles was (1.8 m/s, 2.8 m/s). Compared with the rear section and middle section of the capsule, the axial velocity on the R1, R2, and R3 measuring rings was basically unchanged, while the axial velocity on the R4 and R5 measuring rings increased.

Figure 13 shows the local variation of axial velocity on the same measuring ring.

The distribution of axial velocity on the same measuring ring was basically the same, showing a wave-like variation trend of alternating crest and trough with the change of polar axis, and the axial velocity fluctuated greatly. The change of the axial velocity between the measuring rings decreased gradually with the increase of the radius of the measuring ring, which indicated that the axial velocity tended to be stable along the path. Three troughs and three peaks appeared in the distribution of axial velocity on the same measuring ring, and the location of the alternate change of troughs and peaks also showed certain regularity. Among them, the wave crest was located at the polar position of 0°, 120°, and 240°, while the wave trough was located at the polar position of 60°, 180°, and 300°, and the position where the wave trough occurred coincides with the arrangement position of the three cylindrical support bodies.

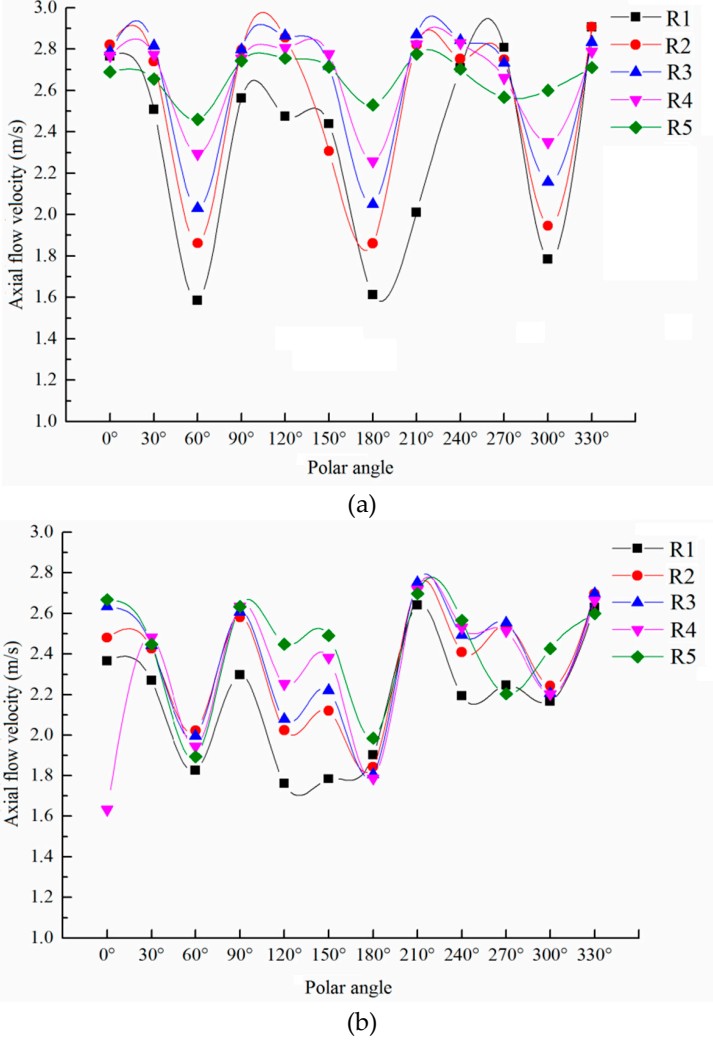

(a)

(b)

**Figure 13.** *Cont.*

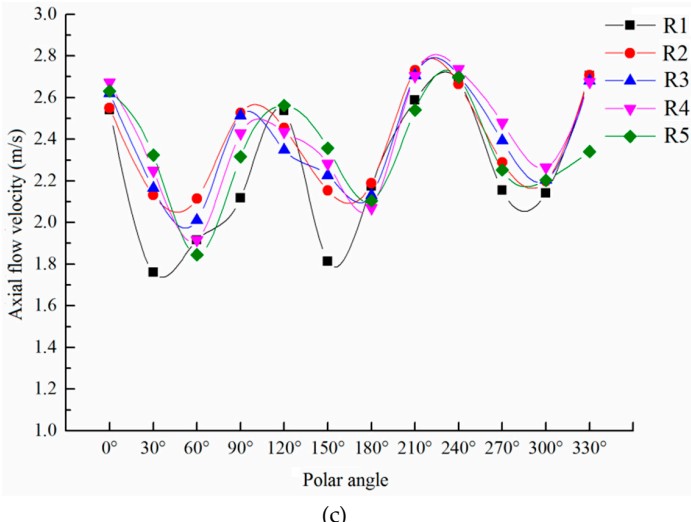

(c)

**Figure 13.** Axial velocity of the same measuring ring: (**a**) rear section; (**b**) middle section; and (**c**) front section.

Figure 14 shows the circumferential flow velocity distribution of the front, middle and rear annular gap sections of the capsule, as follows:

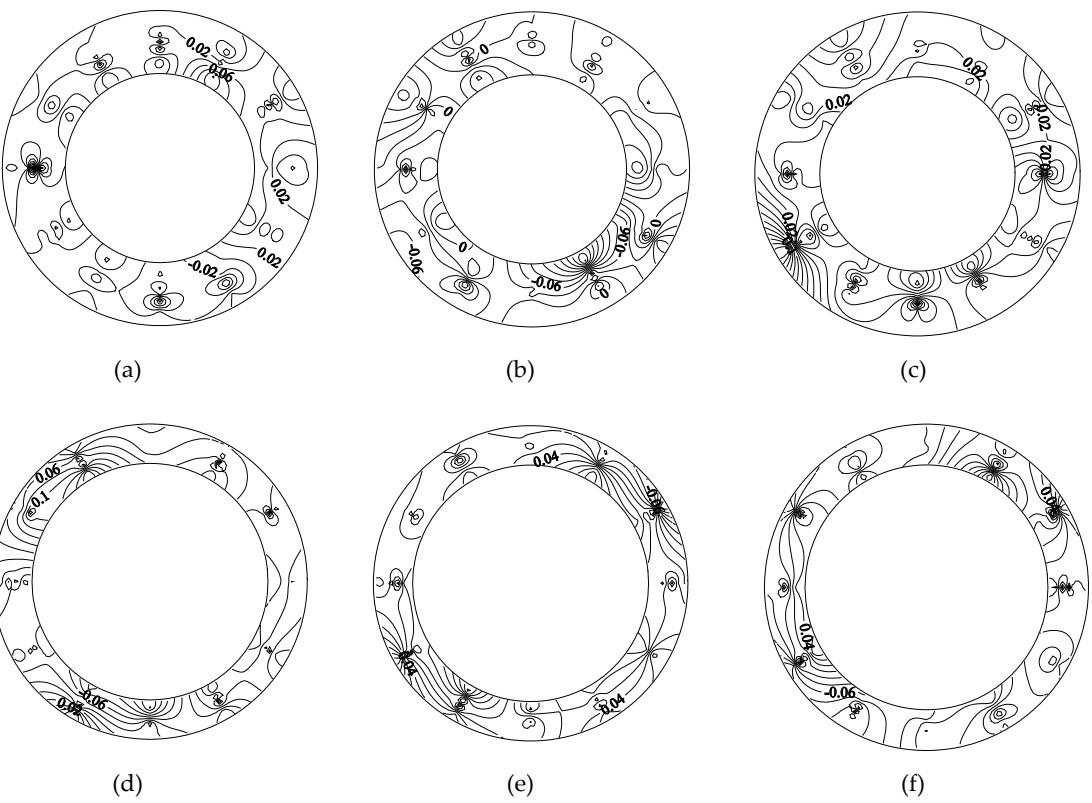

**Figure 14.** Circumferential flow velocity distribution of annular gap sections of the capsules: (**a**) $L/d$ = 2.5, rear section; (**b**) $L/d$ = 2.5, middle section; (**c**) $L/d$ = 2.5, front section; (**d**) $L/d$ = 2.14, rear section; (**e**) $L/d$ = 2.14, middle section; and (**f**) $L/d$ = 2.14, front section.

Along the movement direction of the capsule, the circumferential velocity contour line of annular gap flow changed from dense to sparse to dense, indicating that the circumferential velocity gradient of annular gap flow decreased first and then increased along the length direction of the capsule. When the flow moved through the rear section of the capsule, the circumferential velocity of the annular gap flow in the rear section of the capsule was generally small; about 0.07 m/s. When the water continued to flow to the middle section of the capsule, the circumferential velocity of the annular gap flow in the middle section of the capsule was significantly higher than that in the rear section of the capsule; about 0.1 m/s. The distribution of circumferential velocity was also flatter than that of the rear section of the capsule, and the density of the contour line of circumferential velocity was also lower than that of the rear section of the capsule. When the flow continued to move to the front section of the capsule, due to the influence of the cylindrical support, the density of the circumferential velocity contour in the anterior section of the capsule increased compared with that in the middle section of the capsule, and the circumferential velocity gradient of the anterior section of the capsule increased gradually.

In order to further analyze the circumferential velocity change of annular gap flow during the movement of the capsule, the capsule with $L/d$ = 2.14 as an example was analyzed from the same polar axis and the same measuring ring.

Figure 15 shows the local variation of circumferential velocity on the same polar axis.

During the movement of the capsule, the circumferential velocity on each polar axis of the annular gap section did not change much, and the distribution of circumferential velocity was relatively stable. However, the circumferential velocity in different annular gap sections was different, and the circumferential velocity in the rear section of the capsule was smaller than that in the front section. Along the movement direction of the capsule, the circumferential velocity of the annular gap section gradually increased along the length direction of the capsule, and its variation range and fluctuation amplitude also gradually increased. The main reason was that the existence of the support body of the capsule can promote the circumferential velocity of the flow, which made the circumferential velocity of the annular gap flow and its variation range increase along the path.

Figure 16 shows the local variation of circumferential velocity on the same measuring ring.

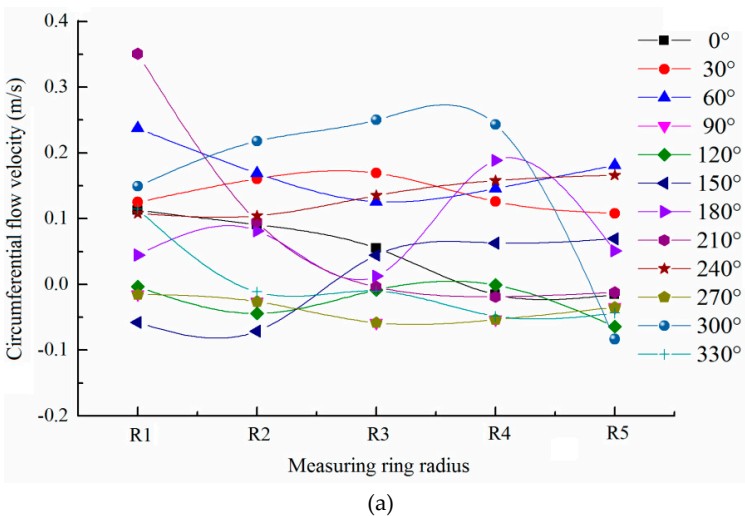

(a)

**Figure 15.** *Cont.*

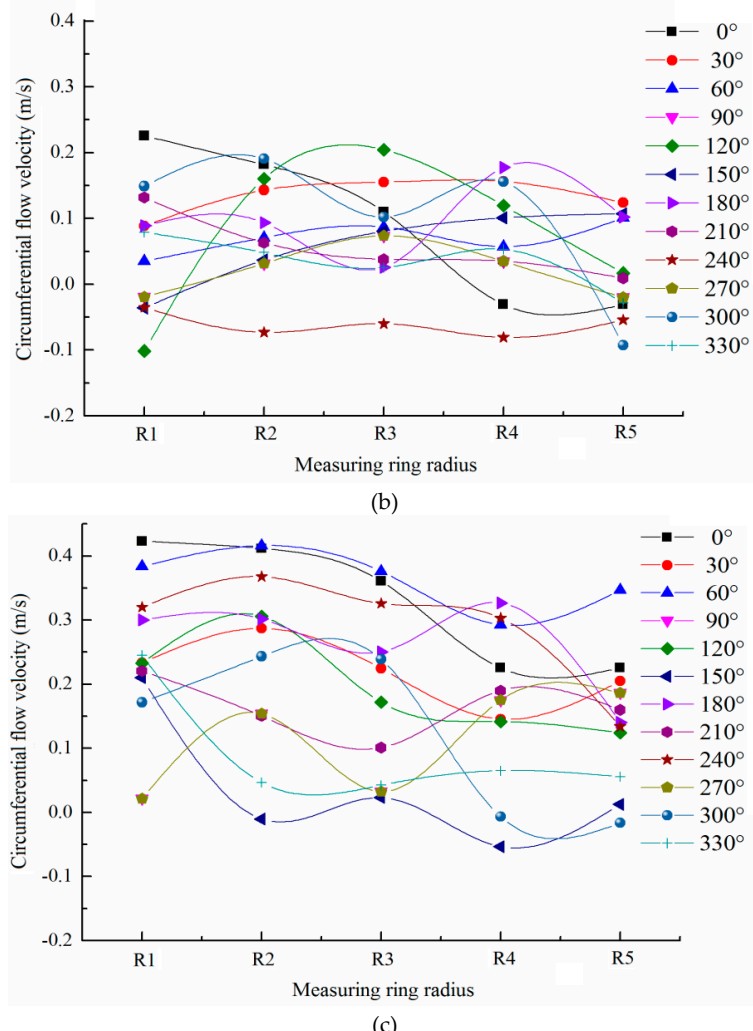

(b)

(c)

**Figure 15.** Circumferential velocity of the same polar axis: (**a**) rear section; (**b**) middle section; and (**c**) front section.

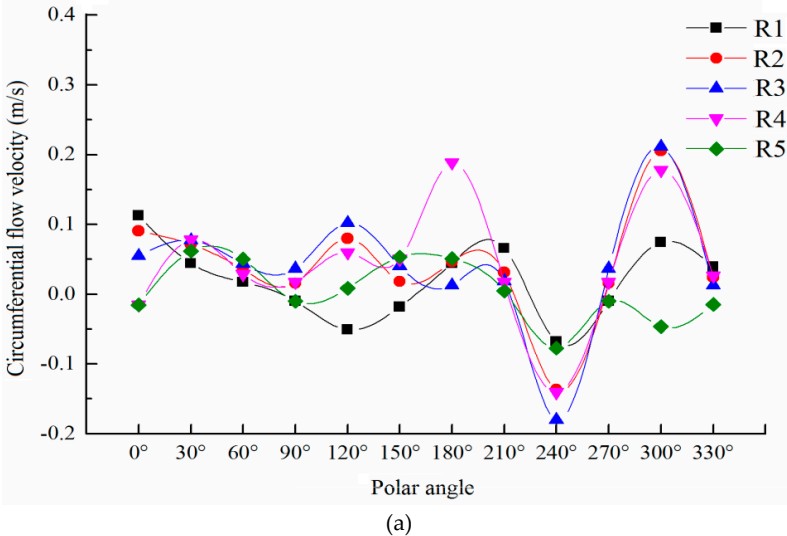

(a)

**Figure 16.** *Cont.*

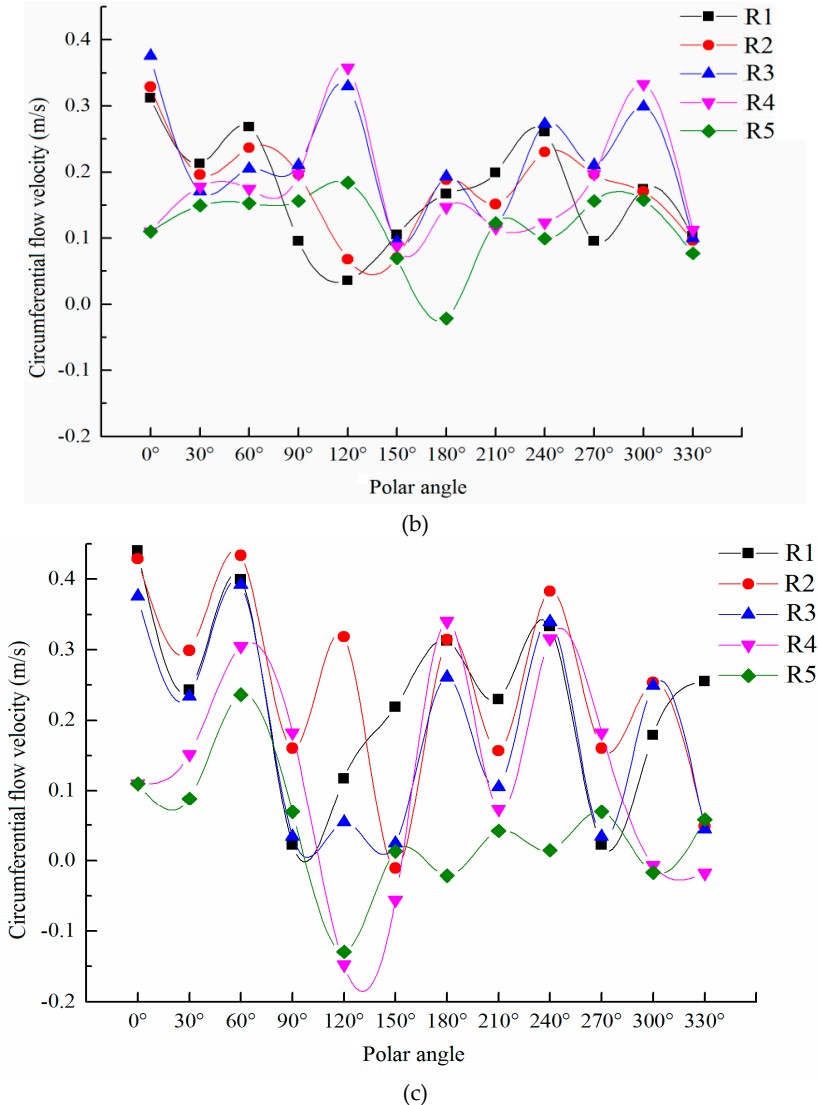

**Figure 16.** Circumferential velocity of the same measuring ring: (**a**) rear section; (**b**) middle section; and (**c**) front section.

Compared with the circumferential velocity distribution of the annular gap flow on the same polar axis, the circumferential velocity distribution of the annular gap flow on the same measuring ring fluctuated greatly, but the fluctuation decreased gradually with the increase of the transport distance. The range width of circumferential velocity of annular gap flow along the length direction of the capsule also decreased from 0.6 to 0.4, indicating that the circumferential velocity of annular gap flow had a gradually concentrated trend along the path. Compared with the axial velocity in the same measuring ring, the alternation of the peak and trough of the circumferential velocity in the same measuring ring was not obvious.

Figure 17 shows the radial flow velocity distribution of the front, middle, and rear annular gap sections of the capsule.

Similar to the circumferential velocity of annular gap flow, the radial velocity contour line of annular gap flow changed from dense to sparse to dense along the moving direction of the capsule, indicating that the radial velocity gradient of annular gap flow decreased first and then increased, and its direction is basically away from the center of the circle. For any annular gap section, the radial velocity of annular gap flow was very small, and its variation range was (−0.16 m/s, 0.38 m/s). Among them, the radial velocity was mostly distributed in the interval (0 m/s, 0.05 m/s) and (0.1 m/s, 0.2 m/s), and the

interval (0.05 m/s, 0.1 m/s) was less. The average radial velocity of annular gap flow was about 0.06 m/s, which was only 1/30 of the average axial velocity of annular gap flow and about 0.7 of the average circumferential velocity of annular gap flow.

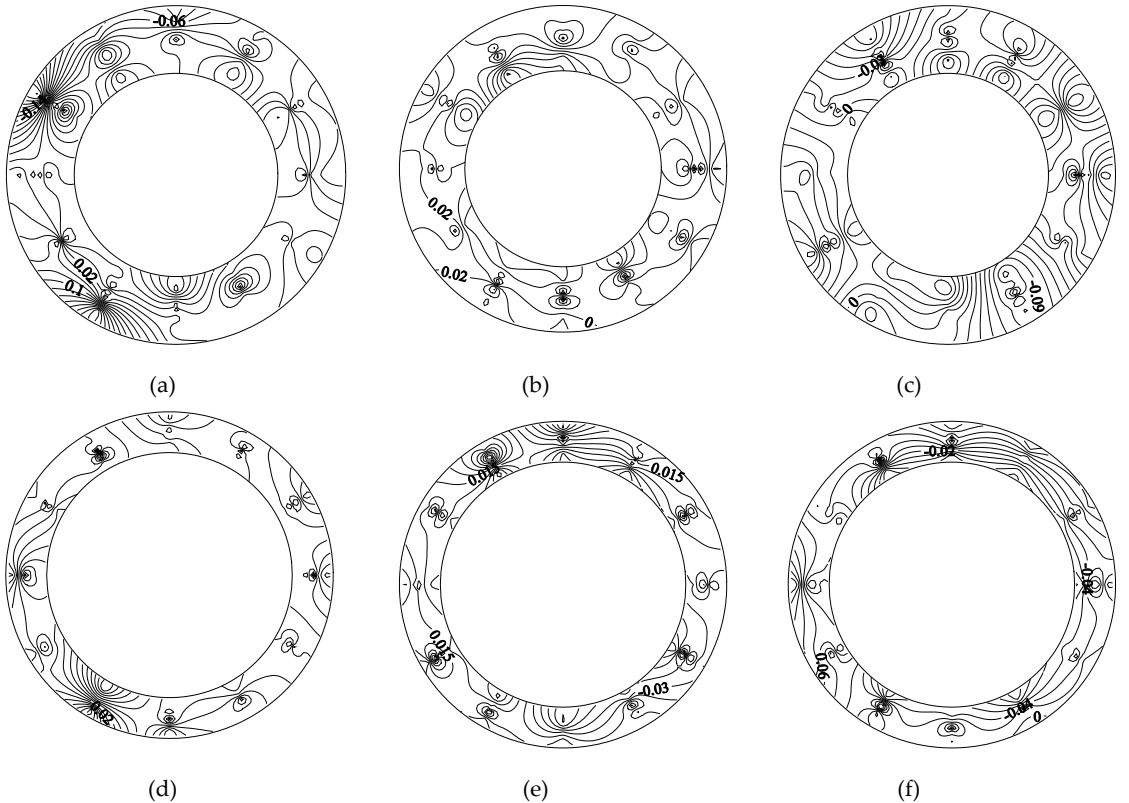

|       |       |       |
|:-----:|:-----:|:-----:|
| (a)   | (b)   | (c)   |
| (d)   | (e)   | (f)   |

**Figure 17.** Radial flow velocity distribution of annular gap sections of the capsules: (**a**) $L/d$ = 2.5, rear section; (**b**) $L/d$ = 2.5, middle section; (**c**) $L/d$ = 2.5, front section; (**d**) $L/d$ = 2.14, rear section; (**e**) $L/d$ = 2.14, middle section; and (**f**) $L/d$ = 2.14, front section.

In order to further analyze the radial velocity change of annular gap flow during the movement of the capsule, the capsule with $L/d$ = 2.14 as an example was analyzed from the same polar axis and the same measuring ring.

Figure 18 shows the local variation of radial velocity on the same polar axis.

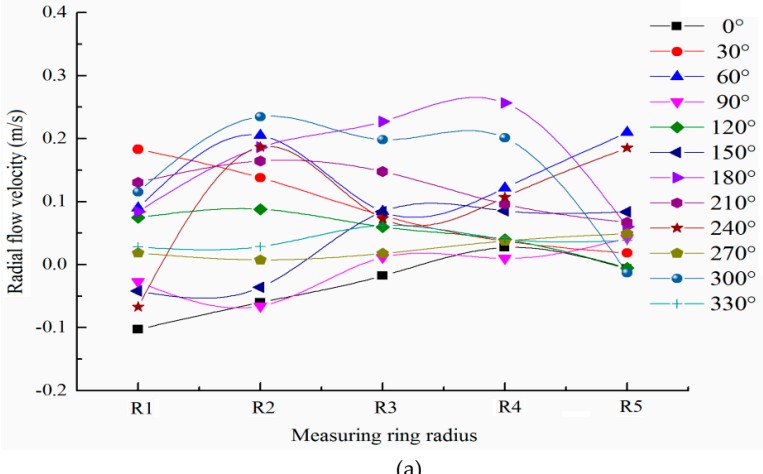

(a)

**Figure 18.** *Cont.*

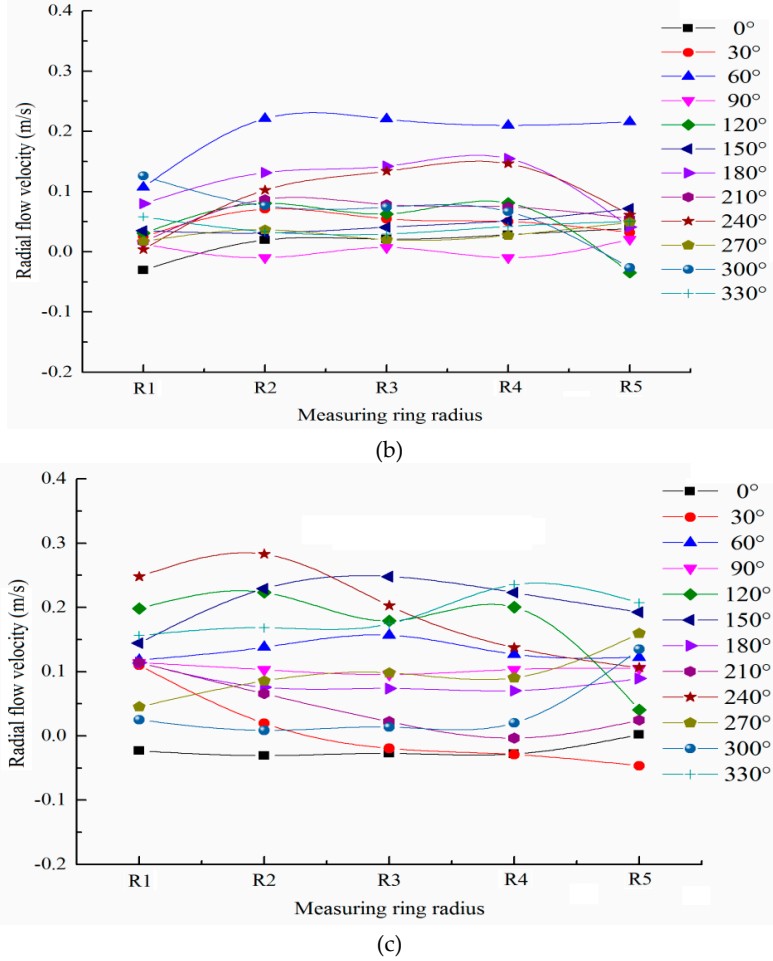

**Figure 18.** Radial velocity of the same polar axis: (**a**) rear section; (**b**) middle section; and (**c**) front section.

Similar to the change of the circumferential velocity on the same polar axis, the radial velocity on each polar axis of the annular gap section changes little and its distribution is relatively stable during the movement of the capsule. When the water flowed around the rear section of the capsule, the radial velocity distribution of the rear section of the capsule was more disordered, and the radial velocity fluctuated greatly for the double influence of the support body and the barrel. When the water continued to flow to the middle section of the capsule, the radial velocity distribution was flatter than that of the rear section of the capsule, and the radial velocity fluctuated less. When the water continued to flow to the front section of the capsule, the radial velocity distribution in the front section of the capsule was more disordered than that in the middle section and its distribution interval was larger than that in the middle section for the influence of the support body of the capsule.

Figure 19 shows the local variation of radial velocity on the same measuring ring.

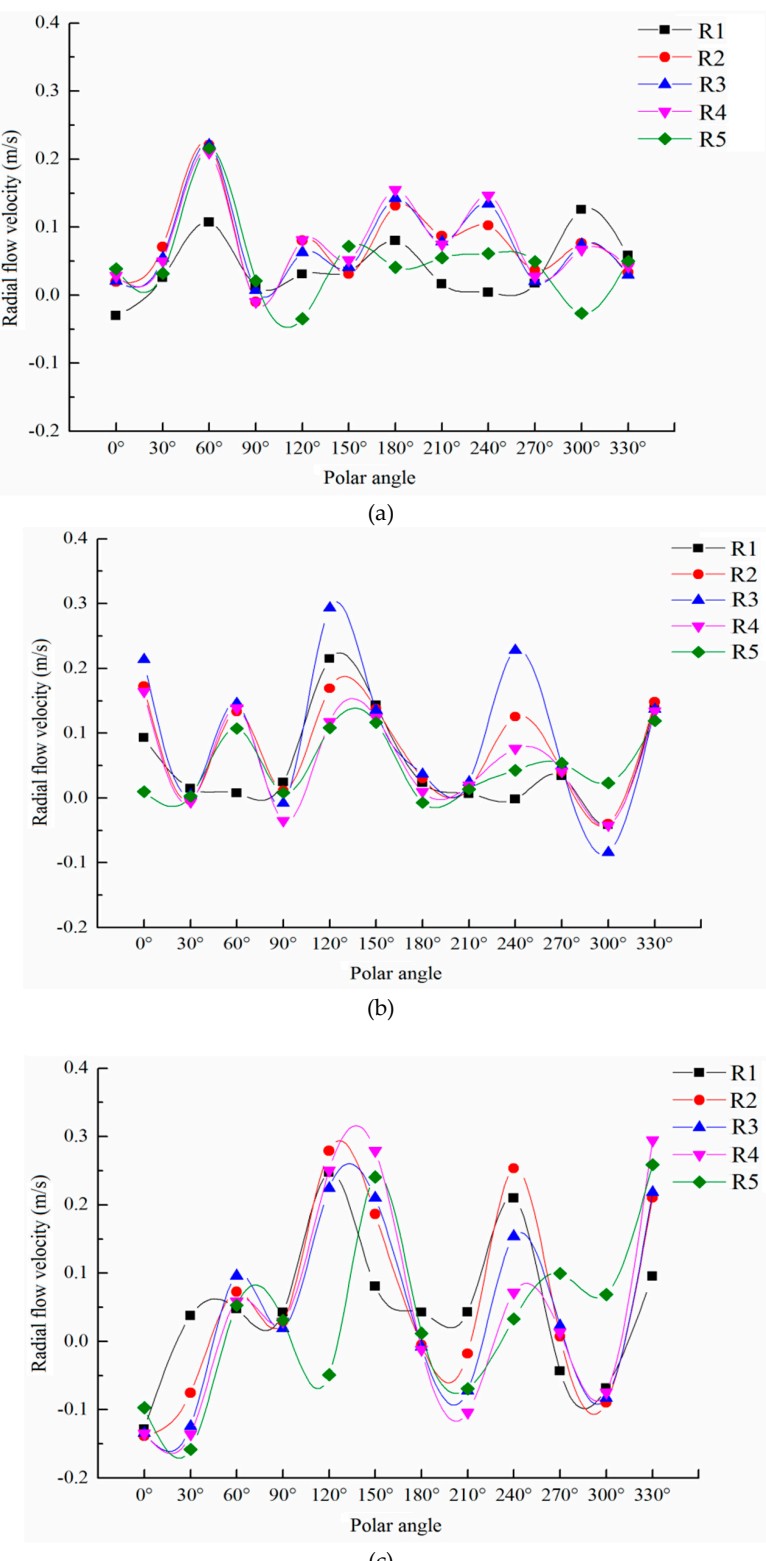

**Figure 19.** Radial velocity of the same measuring ring: (**a**) rear section; (**b**) middle section; and (**c**) front section.

Compared with the radial velocity distribution of the annular gap flow on the same polar axis, the radial velocity distribution of the annular gap flow on the same measuring ring fluctuated greatly, and its fluctuation gradually increased along the length direction of the capsule. Similar to the axial velocity distribution on the same measuring ring, the radial velocity distribution on the same measuring

ring also showed a wave-like variation trend of alternating crest and trough with the change of the polar axis.

## 4. Conclusions

(1) For the cross section near the upstream of the capsule, the axial velocity distribution was smaller near the middle of the pipe and larger near the inner wall of the pipe; the circumferential velocity was distributed around the support body of the capsule, showing a symmetrical distribution; and the radial velocity distribution was more thinly spread near the pipe wall and denser near the center of the pipe.

(2) For the cross section near the downstream of the capsule, the axial velocity distribution was smaller near the middle of the pipe and larger near the inner wall of the pipe; the circumferential velocity was distributed near the support body of the capsule, and the opposite direction of the circumferential velocity was symmetrically distributed in the left and right positions of the support body; and the radial velocity distribution was more thinly spread near the pipe wall and denser near the center of the pipe.

(3) For any annular gap section, the radial velocity of annular gap flow was very small, and the average radial velocity of annular gap flow was about 1/30 of the average axial velocity of annular gap flow and about 0.7 of the average circumferential velocity of annular gap flow. The axial flow velocity, circumferential flow velocity, and radial flow velocity on the same measuring ring showed a wave-like variation trend of alternating crest and trough with the change of the polar axis.

(4) Along the direction of the movement of the capsule, the axial flow velocity gradient of the annular gap flow decreased along the length of the capsule. The circumferential flow velocity gradient and radial flow velocity gradient of annular gap flow first decreased and then increased along the length of the capsule.

(5) In the process of transporting materials, the influence of the capsule on the flow field in the upstream section was less than that in the downstream section.

In this paper, the study provides theoretical basis for further research on transportation energy consumption and the optimization of the scheme in the practical application of the capsule pipeline hydraulic transportation.

**Author Contributions:** Data curation, Y.L., Y.G. and X.Z.; Funding acquisition, X.S.; Investigation, Y.L. and Y.G.; Writing—original draft, Y.L.; Writing—review and editing, X.S. and X.Z. All authors have read and agreed to the published version of the manuscript.

**Funding:** The research was funded by the National Natural Science Foundation of China (51179116, 51109155) and the Natural Science Foundation of Shanxi Province (2015011067, 201701D221137).

**Acknowledgments:** The research was supported by the Collaborative Innovation Center of New Technology of Water-Saving and Secure and Efficient Operation of Long-Distance Water Transfer Project at Taiyuan University of Technology.

**Conflicts of Interest:** The authors declare that they have no conflicts of interest with respect to the research, authorship, and publication of this article.

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
