# Peer review of "Study on Flow Velocity during Wheeled Capsule Hydraulic Transportation in a Horizontal Pipe"

_water, doi:10.3390/w12041181_

Round 1
Reviewer 1 Report
Comments and Suggestions for Authors
The authors need to address the following comments:
- The manuscript title required modification: Study on hydraulic characteristics of ...what? (I GUESS ''FLOW VELOCITY'') during the......
- The methodology is missing in the abstract. The authors only mention wheeled capsule with the length-diameter ratio and pipeline diameter. The abstract requires modification to include research methodology.
- The full meaning of the PDA needs to be stated before the abbreviation (see Section 2. Experimental Design ).
- The title and labels of figure 1 is not visible. I encourage the authors to re-write them.
- Figure 2(a) is not visible. I encourage the authors to re-design it.
Secondly, the label ''(a)'' for figure 2 need to come before the design details. - Figure 3 and Figure 4 are not visible. It is also difficult to understand the labels of the figures. I encourage the author to revisit the figures and do a proper amendment.
- What is the implication of the results in section 3.1. ''The flow velocity distribution of the upstream section of the capsule ''? Does it support/against any theory or previous study?
- Figure labels and details of most of the figures were poorly written. For example details of Figure 3, 4 6 etc., are difficult to understand. The authors must modify them accordingly.
-
Figure 12 labels are very tiny. I recommend the labels and legend size to be increasing bold accordingly. Same thing for the figure 13, 14, 15, 16, 18 and 19.
How is this finding, particularly, Section 3.3. ( Flow velocity characteristics of annular gap section between capsule and pipe) support/against the previous study or any theory?
- Figure 19 should come before the conclusion or move to the appendix
Author Response
Dear editors and reviewers:
The serial number of this manuscript is water-762338, which is titled “Study on hydraulic characteristics of during the wheeled capsule hydraulic transportation in a horizontal pipe”. First of all, we are grateful for your valuable review comments on this manuscript. Your review comments precisely pointed out several deficiencies in this manuscript, which has profound guiding significance for further modifications and improvements of this manuscript and our future research work. We will give detailed answers to the above review comments proposed by the editors and reviewers one by one below.
- The manuscript title required modification: Study on hydraulic characteristics of ...what? (I GUESS ''FLOW VELOCITY'') during the......
Answer:We have revised “the manuscript title” into “Study on flow velocity during the wheeled capsule hydraulic transportation in a horizontal pipe”.
- The methodology is missing in the abstract. The authors only mention wheeled capsule with the length-diameter ratio and pipeline diameter. The abstract requires modification to include research methodology.
Answer:We have added research methodology in the abstract and the revision can be seen in Line 11-12 of Page1. - The full meaning of the PDA needs to be stated before the abbreviation (see Section 2. Experimental Design ).
Answer:We have added the full meaning of the PDA and the revision can be seen in Line 125-126 of Page3.
- The title and labels of figure 1 is not visible. I encourage the authors to re-write them.
Answer:We have modified it and the revision can be seen in Figure 1 of Page3.
- Figure 2(a) is not visible. I encourage the authors to re-design it.
Secondly, the label ''(a)'' for figure 2 need to come before the design details.
Answer:We have modified it and the revision can be seen in Figure 2 of Page4.
- Figure 3 and Figure 4 are not visible. It is also difficult to understand the labels of the figures. I encourage the author to revisit the figures and do a proper amendment.
Answer:We have modified it and the revision can be seen in Figure 3 of Page4 and Figure 4of Page5.
- What is the implication of the results in section 3.1. ''The flow velocity distribution of the upstream section of the capsule ''? Does it support/against any theory or previous study?
Answer:We have modified the part and the revision can be seen in Line173-179 of Page 5.
8 .Figure labels and details of most of the figures were poorly written. For example details of Figure 3, 4 6 etc., are difficult to understand. The authors must modify them accordingly. Figure 12 labels are very tiny. I recommend the labels and legend size to be increasing bold accordingly. Same thing for the figure 13, 14, 15, 16, 18 and 19.
Answer:We have modified these figures and the revision can be seen in Figure 3-19.
- How is this finding, particularly, Section 3.3. ( Flow velocity characteristics of annular gap section between capsule and pipe) support/against the previous study or any theory?
Answer:We have modified the part and the revision can be seen in Line173-179 of Page 5.
- Figure 19 should come before the conclusion or move to the appendix
Answer:We have modified it and the revision can be seen in Figure 19 of Page21-22.
We have adopted red fonts to highlight the revised parts of the manuscript, which will help the editors and reviewers to review the manuscript again.
We have already answered review comments put forward by the editors and reviewers one by one in detail, and improvements and modifications have been made in the corresponding positions of this manuscript. We hope that the editors and reviewers will review the revised manuscript again. Thanks again to the editors and reviewers for their valuable review comments. If there are still any deficiencies in this manuscript, please don’t hesitate to contact me at the address below, and we will actively cooperate with the editors and reviewers to promptly modify the deficiencies in the manuscript. We deeply appreciate your consideration of our manuscript, and we are looking forward to receiving comments from the editors and reviewers.
Thank you and best regards,
Yours sincerely,
Li Yongye
The first author:
Name: Li Yongye
E-mail: liyongye@tyut.edu.cn
Phone: +8613934239832
Address: College of Water Resource Science and Engineering, Taiyuan University of Technology, No. 79, Yingze Street, Wanbailin District, Taiyuan 030024, PR China.
Reviewer 2 Report
I want to state the following:
- I think that some minor flaws should be corrected.
- I have the following comments to the authors:
1) To improve the understanding of your experiment, it will be very interesting to state capsule velocity vs. hydraulic floe velocity.
2) Your election of front section and rear section is strange. In English the rear section of a rocket is the motor and the front section the nose cone. So it will be interesting that you state where are located upstream and downstream.
3) Some figures are perfect, others need improvement because low resolution paste. Numbers can not be read, or some foot are half missing.
Lines 184-185. I would replaced "large" with " more thinly spread".
Line 219 Three capsules?
Lines 230-231 I do not see that in Fig.10. Perhaps you want to state: radial velocity distribution...(See lines 184-185 comment
Lines 366-367 I do not agree to radial velocity comment. Perhaps some words (like gradient, distributions, ...) are missing
Line 371 Same problem
Line 373-375. Looking at figures 12a, 15a, 16a, I do not see you conclusion
Author Response
Modification instructions
Dear editors and reviewers:
The serial number of this manuscript is water-762338, which is titled “Study on hydraulic characteristics of during the wheeled capsule hydraulic transportation in a horizontal pipe”. First of all, we are grateful for your valuable review comments on this manuscript. Your review comments precisely pointed out several deficiencies in this manuscript, which has profound guiding significance for further modifications and improvements of this manuscript and our future research work. We will give detailed answers to the above review comments proposed by the editors and reviewers one by one below.
1) To improve the understanding of your experiment, it will be very interesting to state capsule velocity vs. hydraulic floe velocity.
Answer:We have added the part and the revision can be seen in Line 146-159 of Page5.
2) Your election of front section and rear section is strange. In English the rear section of a rocket is the motor and the front section the nose cone. So it will be interesting that you state where are located upstream and downstream.
Answer:We have modified it and the revision can be seen in Line 168-169 of Page5 and Figure4.
3) Some figures are perfect, others need improvement because low resolution paste. Numbers can not be read, or some foot are half missing.
Lines 184-185. I would replaced "large" with " more thinly spread".
Answer:We have modified it and the revision can be seen in Line 231-232 of Page8.
Line 219 Three capsules?
Answer:We have modified it and the revision can be seen in Line 291 of Page10.
Lines 230-231 I do not see that in Fig.10. Perhaps you want to state: radial velocity distribution...(See lines 184-185 comment
Answer:We have modified it and the revision can be seen in Line 305-307 of Page11.
Lines 366-367 I do not agree to radial velocity comment. Perhaps some words (like gradient, distributions, ...) are missing
Line 371 Same problem
Answer:We have modified the part and the revision can be seen in Line 520-521 of Page22 and Line 526-527 of Page22.
Line 373-375. Looking at figures 12a, 15a, 16a, I do not see you conclusion
Answer:We have modified it and the revision can be seen in Line528-530 of Page22.
We have adopted red fonts to highlight the revised parts of the manuscript, which will help the editors and reviewers to review the manuscript again.
We have already answered review comments put forward by the editors and reviewers one by one in detail, and improvements and modifications have been made in the corresponding positions of this manuscript. We hope that the editors and reviewers will review the revised manuscript again. Thanks again to the editors and reviewers for their valuable review comments. If there are still any deficiencies in this manuscript, please don’t hesitate to contact me at the address below, and we will actively cooperate with the editors and reviewers to promptly modify the deficiencies in the manuscript. We deeply appreciate your consideration of our manuscript, and we are looking forward to receiving comments from the editors and reviewers.
Thank you and best regards,
Yours sincerely,
Li Yongye
The first author:
Name: Li Yongye
E-mail: liyongye@tyut.edu.cn
Phone: +8613934239832
Address: College of Water Resource Science and Engineering, Taiyuan University of Technology, No. 79, Yingze Street, Wanbailin District, Taiyuan 030024, PR China.
Reviewer 3 Report
In the manuscript, not all drawings are visible and captions are not legible.
Experiment object (Figure1) - wheeled capsule and experimental device (Figure 2) have already been presented in other publications.
The text shows that the main task was to "measure the flow velocity by PDA in the pipeline during the movement of the capsule in the test section" (see section 3).
An important issue regarding the PDA (Phase Doppler Anemometer) measurement condition, such as light scattering, scattering angle, particle velocity, particle size, detectors, has been completely omitted.
The local flow velocity field for capsule flow in the hydraulic pipeline was not analyzed.
The main purpose of the analysis should be to establish the dependence of local flow parameters, such as pressure and velocity.
The CFD predictions of flow velocity fields should be verified during measurements.
It is not known what the benefits of these studies are for the movement of wheeled capsules in a hydraulic capsule pipeline (HCPs).
Author Response
Modification instructions
Dear editors and reviewers:
The serial number of this manuscript is water-762338, which is titled “Study on hydraulic characteristics of during the wheeled capsule hydraulic transportation in a horizontal pipe”. First of all, we are grateful for your valuable review comments on this manuscript. Your review comments precisely pointed out several deficiencies in this manuscript, which has profound guiding significance for further modifications and improvements of this manuscript and our future research work. We will give detailed answers to the above review comments proposed by the editors and reviewers one by one below.
1)In the manuscript, not all drawings are visible and captions are not legible.
Answer:We have revised all the invisible drawings in the manuscript.
2)Experiment object (Figure1) - wheeled capsule and experimental device (Figure 2) have already been presented in other publications.
Answer:Our research is carried out under the same experimental device. Previously published papers mainly focused on the motion characteristics of the capsule. This paper mainly studied the flow velocity characteristics of the flow in the process of the movement of the capsule. Experiment object (Figure 1) and wheel capsule and experimental device (Figure 2) are cited in the paper.
3) The text shows that the main task was to "measure the flow velocity by PDA in the pipeline during the movement of the capsule in the test section" (see section 3). An important issue regarding the PDA (Phase Doppler Anemometer) measurement condition, such as light scattering, scattering angle, particle velocity, particle size, detectors, has been completely omitted.
Answer:The flow velocity is measured by Phase Doppler Analyzer (PDA) in the pipeline during the movement of the capsule in the test section. In order to prevent the refraction of the pipeline surface to the laser, a rectangular water jacket is installed outside the test pipeline, as shown in Fig. 2b. The particle size of the tracer used in the experiment is 30 μ m.
4) The local flow velocity field for capsule flow in the hydraulic pipeline was not analyzed. The main purpose of the analysis should be to establish the dependence of local flow parameters, such as pressure and velocity.
Answer:In this paper, we mainly study the distribution of the whole velocity flow field during the movement of the capsule. Next, we will study the local flow field and establish the dependence of local flow parameters, such as pressure and velocity.
5) The CFD predictions of flow velocity fields should be verified during measurements.
In this paper, the flow velocity in the pipe during the movement of the capsule is mainly studied by experiments, without involving CFD.
6) It is not known what the benefits of these studies are for the movement of wheeled capsules in a hydraulic capsule pipeline (HCPs).
Answer:The study of this paper provides theoretical basis for further research on the transportation energy consumption and the optimization of the scheme in the practical application of the capsule pipeline hydraulic transportation.
We have adopted red fonts to highlight the revised parts of the manuscript, which will help the editors and reviewers to review the manuscript again.
We have already answered review comments put forward by the editors and reviewers one by one in detail, and improvements and modifications have been made in the corresponding positions of this manuscript. We hope that the editors and reviewers will review the revised manuscript again. Thanks again to the editors and reviewers for their valuable review comments. If there are still any deficiencies in this manuscript, please don’t hesitate to contact me at the address below, and we will actively cooperate with the editors and reviewers to promptly modify the deficiencies in the manuscript. We deeply appreciate your consideration of our manuscript, and we are looking forward to receiving comments from the editors and reviewers.
Thank you and best regards,
Yours sincerely,
Li Yongye
The first author:
Name: Li Yongye
E-mail: liyongye@tyut.edu.cn
Phone: +8613934239832
Address: College of Water Resource Science and Engineering, Taiyuan University of Technology, No. 79, Yingze Street, Wanbailin District, Taiyuan 030024, PR China.
Round 2
Reviewer 3 Report
The authors comprehensively answered the reviewer's questions and doubts.
The manuscript text has been significantly improved.
It can be considered that the study of this paper provides the theoretical basis for further research on the practical application of the wheeled capsules hydraulic transportation.